# Estimating Soil Properties and Nutrients by Visible and Infrared Diffuse Reflectance Spectroscopy to Characterize Vineyards

**José Ramón Rodríguez-Pérez** [1,*] , **Víctor Marcelo** [2] , **Dimas Pereira-Obaya** [1] , **Marta García-Fernández** [1] **and Enoc Sanz-Ablanedo** [1]

1   Grupo de Investigación en Geomática e Ingeniería Cartográfica (GEOINCA), Universidad de León, Avenida de Astorga s/n, 24401 Ponferrada, León, Spain; dperep01@estudiantes.unileon.es (D.P.-O.); mgarcf@unileon.es (M.G.-F.); esana@unileon.es (E.S.-A.)
2   Departamento de Ingeniería y Ciencias Agrarias, Universidad de León, Avenida de Astorga s/n, 24401 Ponferrada, León, Spain; v.marcelo@unileon.es
*   Correspondence: jr.rodriguez@unileon.es; Tel.: +34-98-744-2000

**Abstract:** Visible, near, and shortwave infrared (VIS-NIR-SWIR) reflectance spectroscopy, a cost-effective and rapid means of characterizing soils, was used to predict soil sample properties for four vineyards (central and north-western Spain). Sieved and air-dried samples were measured using a portable spectroradiometer (350–2500 nm) and compared for pistol grip (PG) versus contact probe (CP) setups. Raw data processed using standard normal variate (SVN) and detrending transformation (DT) were grouped into four subsets (VIS: 350–700 nm; NIR: 701–1000 nm; SWIR: 1001–2500 nm; and full range: 350–2500 nm) in order to identify the most suitable range for determining soil characteristics. The performance of partial least squares regression (PLSR) models in predicting soil properties from reflectance spectra was evaluated by cross-validation. The four spectral subsets and transformed reflectances for each setup were used as PLSR predictor variables. The best performing PLSR models were obtained for pH, electrical conductivity, and phosphorous ($R^2$ values above 0.92), while models for sand, nitrogen, and potassium showed moderately good performances ($R^2$ values between 0.69 and 0.77). The SWIR subset and SVN + DT processing yielded the best PLSR models for both the PG and CP setups. VIS-NIR-SWIR reflectance spectroscopy shows promise as a technique for characterizing vineyard soils for precision viticulture purposes. Further studies will be carried out to corroborate our findings.

**Keywords:** spectroscopy; PLSR; pistol grip; contact probe; vineyard soils



## 1. Introduction

Knowledge of soil properties and mapping is regarded as key to decision making in precision viticulture, mainly because of a growing interest in more environmentally friendly and sustainable practices [1]. Chemical and physical characteristics are especially important in evaluating soil fertility and understanding soil dynamics [2]. Modern viticulture requires the evaluation of a wide range of soil properties in a timely and cost-effective way. However, conventional methods for laboratory analysis of soils are expensive, time-consuming, and non-environmentally friendly (they require the use of chemical reagents), and need a whole range of sophisticated protocols and equipment [3]. Soil assessment using visible (VIS), near infrared (NIR), and shortwave infrared (SWIR) spectroscopy, although it cannot replace laboratory chemical analysis, is fast, cost-effective, environmentally friendly, non-destructive, reproducible, and repeatable analytical technique [4]. It is also easy to use since samples only require minimal preparation, and, furthermore, it requires no chemicals or reagents and so does not generate chemical waste [5]. A single wavelength spectrum may contain comprehensive information that can predict various soil components [6]. Spectroscopic applications to the soil include NIR, VIS-NIR, and mid-infrared (MIR) analyses

comprising Fourier transform infrared (FTIR), FTIR-attenuated total reflection (FTIR-ATR), and Raman spectroscopy [3].

Spectroscopic techniques are physical characterisation methods that involve studying electromagnetic wave interaction with the material under consideration in the ultraviolet, VIS, and infrared (IR) wavelengths [7]. Furthermore, spectroscopy, when coupled with multivariate data analysis, has been shown to be a powerful tool for developing quantitative and classification models in many disciplines, including food technology [8], petroleum engineering [9], and soil science [10], as described by Barra et al. [3]. VIS-NIR spectroscopy is an empirical method based on an analysis of diffuse reflectance radiation in relation to a material's characteristics and the assumption that the concentration of a given constituent is a linear combination of several absorption features [11].

Ben-Dor [12] described the principles and mechanisms of soil–radiation interactions in relation to quantitative remote sensing of soil properties, noting problematic factors that prevent direct spectral analysis of electromagnetic signals and reviewing studies that describe advances in this quantitative method. The same author previously published research focused on the reflectance spectrum in the VIS-NIR-SWIR regions, together with proposals for practical applications [13]. Stenberg et al. [14] comprehensively reviewed the literature on soil VIS and IR diffuse reflectance spectroscopy (including fundamentals, studied soil properties, conditioning factors, calibrations, field analyses, and practical applications), while Kuang et al. [15] reviewed the sensing concept applied to soil properties (basics and brief theory, factors affecting results, and relationship between sensor output and soil properties).

When electromagnetic radiation is directed to a soil sample, it causes individual molecular bonds to vibrate (they bend or stretch), resulting in a characteristic absorption spectrum [15]. The resulting spectrum has a specific shape dependent on soil composition that can be used for physical and chemical analyses [14]. Soil content in carbon (C), nitrogen (N), water, and clay minerals are properties with direct NIR spectral responses that can be attributed to overtones of OH and overtones and/or combinations of C-H + C-H, C-H + C-C, $OH^+$ minerals, and N-H. Moreover, absorption bands in the VIS range (400–700 nm), due to electron excitation, are related to soil colour [15,16]. Numerous studies have used VIS-NIR spectroscopy in an attempt to predict soil content in total and organic C, total N, clay minerals, and water. Other studies have focused specifically on sand and silt content, pH, electrical conductivity (Ec), total content in N, extractable phosphorous (P), extractable potassium (K), extractable calcium (Ca), extractable iron (Fe), extractable sodium (Na), extractable manganese (Mn), extractable magnesium (Mg), and cation exchange capacity (CEC) [16–19]. However, results for those studies have been typically modest and also highly variable, as they were based on co-variations in constituents that are spectrally active.

The availability of commercial spectroscopic equipment and software packages for multivariate calibration has led to VIS-NIR spectroscopy becoming widely used for soil characterisation purposes. Standards and protocols for reflectance measurements of soils in the laboratory have been proposed by Pimstein et al. [20] and Ben-Dor et al. [21], while Kuang et al. [15] have reviewed several studies of different VIS-NIR reflectance sensors, including laboratory, non-mobile/field (in situ), and mobile/field (online) sensors.

Diffuse reflectance spectra in soil are non-specific, since scatter effects caused by structure result in overlapping absorption features. Therefore, multivariate techniques are required to extract absorption patterns and to correlate spectra with soil properties. Calibration methods for soil applications include linear regression approaches, such as stepwise multiple linear regression (SMLR), principal component regression (PCR), and partial least squares regression (PLSR), and also data mining techniques, such as neural networks (NN), multivariate adaptive regression splines (MARS), boosted regression trees (BRT), random forests (RF), and support vector machines (SVM), along with their combinations [14].

In agriculture, quantitative and qualitative analyses of soil properties yield accurate information to guide the management of soil fertility and productivity through adjusted

fertiliser formulations and recommendations [22]. The rapid development of portable and handheld spectrometers allows analyses to be conducted in situ [23]. As a key factor for site-specific management practices, Angelopoulou et al. [24] recently reviewed laboratory and proximal sensing spectroscopy in the VIS, NIR, and SWIR wavelength regions for soil organic matter estimates. MIR spectroscopy and laser diffraction analysis (LDA) have also been demonstrated to be useful for calculating organic matter and clay content in soils [25].

Spectroscopy has previously been applied to viticulture. For vineyards located in Australia, Cozzolino et al. [23] evaluated use of a portable NIR spectrophotometer in the field to predict soil chemical properties, fitting PLSR models with coefficients of determination ($R^2$) that ranged from 0.69 for P to 0.95 for total N content. Muganu et al. [26] demonstrated the great potential of NIR-acoustic optically tuneable filter (AOTF) spectroscopy in assessing grape quality, noting the influence of soil management practices on vine and grape characteristics. Páscoa et al. [27] developed a method for indirect soil differentiation based on grapevine leaf spectra, demonstrating that leaf spectral information can be used to define soil maps for vineyards. For Northern Portugal, Lopo et al. [28] demonstrated the ability of NIR spectroscopy to discriminate between vineyard soil types, showing that water content is not a significant factor in differentiating between soils.

As reported by Marín-González et al. [19], VIS-NIR spectroscopy can be used to detect soil properties using laboratory, in situ, and online measurements. This technique is effective mainly for assessing primary soil properties with direct spectral responses in the VIS-NIR range, e.g., water, C, N, and clay [15], as well as other soil chemical parameters in the laboratory [17]. Few studies, however, have described evaluations of soil properties without direct spectral responses in the VIS-NIR-SWIR range or have compared different approaches to spectral preprocessing and the use of different accessories. Marín-González et al. [19] evaluated models to estimate soil properties without direct spectral responses in the NIR spectroscopy range (CEC, pH, and extractable Ca and Mg), reporting very good accuracy for pH and moderately good accuracy for CEC and Mg. Munnaf et al. [29] explored accuracy improvements to visible NIR spectroscopy estimates of secondary soil properties (pH and extractable K, Mg, Ca and Na) by laboratory fusion approaches, finding that exclusively online spectrum or hybrid models (50% online scanned spectra and laboratory spectra) significantly improved online prediction accuracies. Note, however, that since those works were based on online spectral measurements obtained by specialist industrial-grade instruments mounted in heavy soil-tilling machinery, and so they are not applicable to multi-year crops such as vineyards.

The objectives of this study were (1) to compare spectral signatures of soils as measured in two setups, using a pistol grip (PG) and fibre optic cable, with light provided by an external illuminator lamp, and using a contact probe (CP), with light provided by an internal halogen bulb; and (2) to assess the ability of linear regression models to calculate soil properties (mainly without direct spectral responses in the VIS-NIR-SWIR range) from preprocessed and non-preprocessed spectral data. Thus, two measurement methods (PG and CP) and two modelling approaches (with and without preprocessing) were applied and compared in order to define a suitable protocol to predict vineyard soil composition by VIS-NIR-SWIR spectroscopy.

## 2. Materials and Methods

### 2.1. Study Area and Soil Sampling

Soils were sampled in four different commercial vineyards belonging to three Designations of Origin (DOs): Bierzo (northwest Spain), Ribera del Duero (north-central Spain), and Rueda (northwest-central Spain). A total of 12 soil samples were collected from each of the vineyards, yielding 48 samples in total. Table 1 summarises the main characteristics of the sampled sites, which were very diverse in terms of soil textures, crops, and landscapes.

**Table 1.** Sites sampled for soils.

| Municipality | Designation of Origin | Grape Cultivar | Longitude | Latitude | Soil Classification |
|---|---|---|---|---|---|
| Cacabelos | Bierzo | Mencía | 6.754 W | 42.626 N | Dystric Cambisol |
| Camponaraya | Bierzo | Godello | 6.692 W | 42.606 N | Chromic Cambisol |
| Valbuena de Duero | Ribera de Duero | Tempranillo | 4.391 W | 41.631 N | Lithic Leptosol |
| Matapozuelos | Rueda | Verdejo | 4.765 W | 41.364 N | Albic Arenosol |

Geographic coordinates refer to WGS84. The soil classification system is that of the IUSS Working Group WRB [30].

Soil samples were collected in the 0–0.40 m layer between June and August 2015. Soil cores were air-dried and were sieved (10-mesh) by hand selecting fractions <2 mm before chemical analyses, performed in the Instrumental Techniques Laboratory attached to León University (certified by UNE-EN ISO 9001). The following official analytical measurement methods [31] were used: particle-size distribution of clay, silt, and sand (%) by the pipette method, pH at 1:2.5 soil/water suspension, Ec (dS m$^{-1}$) at 1:5 soil/water suspension, organic matter (%) by the Walkley–Black method, N (%) by total Kjeldahl nitrogen, P extracted with $NaHCO_3$ 0.5 M at pH 8.5 by optical spectrometer UV/VIS analysis (mg kg$^{-1}$), K and Ca extracted with $AcONH_4$ 1N at pH 7 by ICP-AES analysis (cmol kg$^{-1}$), Mn and Fe extracted with DTPA at pH 7.3 by ICP-AES analysis (mg kg$^{-1}$), and CEC measured by extraction with $ClBa$ 0.1 M by ICP-AES analysis (cmol kg$^{-1}$).

## 2.2. Spectral Reflectance Acquisition

Soil samples were air-dried and spread in black soil cores (20 × 20 cm). Spectral reflectances were recorded at 1 nm intervals from 350 nm to 2500 nm using an ASD Field-Spec 4 Portable Spectroradiometer (Analytical Spectral Devices, Inc., Boulder, CO, USA). Measurements were made, using a 1.5 m fibre optic cable (25° field-of-view), in two ways: (1) PG setup, with two tungsten halogen lamps supporting the fibre optic; and (2) CP setup, with an internal halogen bulb attached by cable.

Data were collected following spectroradiometer manufacturer recommendations [32]. Spectral measurements corresponded to reflectance calculated as the ratio of reflected soil sample energy to reflected energy of a reference calibration panel, consisting of a white reflectance panel providing a diffuse homogeneous mix of full-source energy at nearly 100%. Recalibration was performed after each measurement of five soil samples.

### 2.2.1. PG Setup Measurements

The geometry parameters of measurements (lamp to soil sample and fibre optic to soil sample distances, and the angle between those two distances) were set to ensure homogenous illumination, with the spot area over the sample surface. To ensure a representative spectrum for each soil sample, four reflectance readings (turning the soil core 90° clockwise before each capture) were calculated, each representing the average of 15 individual measurements.

### 2.2.2. CP Setup Measurements

The CP accessory with an internal halogen bulb allowed the fibre optic to be attached at a fixed measurement angle of 35°, reducing noise caused by shadows and other errors associated with stray light [33]. The sensed spot had a diameter of 10 mm, so measurements were made five times at five different points of the samples and then averaged.

### 2.2.3. Preprocessing

The spectral signatures were preprocessed to identify outliers, and the spectra measured for each sample were averaged. To identify the most suitable range to estimate soil

properties, wavelengths were grouped into four spectral subsets: VIS (350–700 nm), NIR (701–1000 nm), SWIR (1001–2500 nm), and full range (350–2500 nm).

Standard normal variate (SVN) and detrending transformation (DT) were used for scatter correction following previous studies of soil composition estimation by spectroscopy [23,34]. SVN removes multiplicative interferences of scatter and particle size effects from spectral data by centring and scaling each spectral signature [34]. DT removes nonlinear trends in spectroscopic data by calculating a baseline function as the least squares fit of a polynomial to the sample spectrum [34].

### 2.2.4. Soil Property Estimation by PLSR

We used PLSR to estimate soil properties (predicted variables) from spectral signatures (predictor variables), given that (as explained above) diffuse reflectance spectra are correlated with soil properties. Since soil spectra show an overlap of weak overtones and combinations of fundamental vibrational bands, multivariate calibration methods were required to quantitatively determine soil properties [35]. PLSR is a generalisation of linear multiple regression that reduces a large number of collinear variables (e.g., reflectance values) to a few non-correlated hidden (latent) variables or factors (see Geladi and Kowalski [36] and Wold et al. [37] for comprehensive descriptions of PLSR).

We fitted several models in order to identify the most suitable procedure. The three reflectance datasets considered were non-preprocessed data and SVN and DT processed data. Additionally, in order to fit simpler and more effective models, an independent model was fitted for each dataset considering the following subsets as independent variables in the PLSR: VIS (350–700 nm), NIR (701–1000 nm), SWIR (1001–2500 nm), and the full range (350–2500 nm).

The resulting models were compared regarding requirements to fit a robust PLSR model: a small number of factors, small errors in leave-one-out cross-validation (CV), and a high $R^2$ [38]. Because of the small number of soil samples, we used the leave-one-out CV procedure to validate the regression models. $R^2$ and root mean square error (RMSE) values for CV were calculated to test the prediction accuracy of each model; also calculated for CV were standard error (SE) values. The ratio of performance to deviation (RPD), i.e., the standard deviation (SD) to SE ratio, was used to test the usability of the calibrated models [38], with an RPD value of 2 or more considered appropriate for soil analysis by spectroscopy [35]. Statistics were calculated according to the following expressions:

$$R^2 = \left( \frac{n(\sum xy) - (\sum x)(\sum y)}{\sqrt{[(n\sum x^2 - (\sum x)^2)(n\sum y^2 - (\sum y)^2)]}} \right)^2 \tag{1}$$

where $y$ is the predicted values, $z$ is the measured values, and $n$ is the number of samples;

$$RMSE = \sqrt{\frac{\sum(y-z)^2}{n}} \tag{2}$$

where $y$ is the predicted values, $z$ is the measured values, and $n$ is the number of samples;

$$SE = \sqrt{\frac{\sum(y-z)^2}{n-1}} \tag{3}$$

where $y$ is the predicted values, $z$ is the measured values, $n$ is the number of samples; and

$$RPD = \frac{SD}{SE} \tag{4}$$

where SD reflects the SD values of the measured variable, $y$ is the predicted values, $z$ is the measured values, and $n$ is the number of samples.

The PLSR factors used in the models were selected on the basis of the lowest RMSE and highest $R^2$ [39]. The criterion to choose the optimal number of factors was based on RMSE and the explained variance of the model: another factor was added to the model if the RMSE was reduced by >2% and the explained variance increased. The maximum number of factors ultimately selected was seven.

## 3. Results

### 3.1. Soil Reflectance Spectra

Soil spectra were mainly dominated by combinations of fundamental vibrational bands for H-C, H-N, and H-O bonds and by weak overtones, especially from the MIR region [35]. The range of reflectance values for the sampled soils and average spectral signatures for the PG and CP setups are shown in Figure 1.

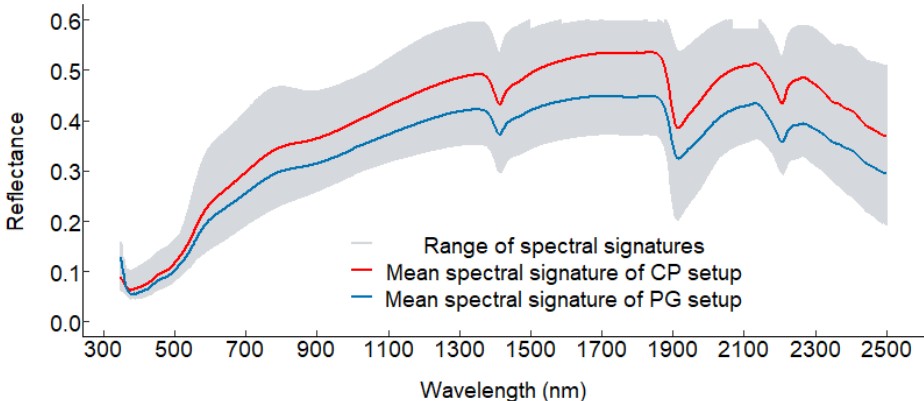

**Figure 1.** Range of reflectance values for the sampled soils and average spectral signatures for the PG and CP setups.

As was expected, the spectral signatures derived from the PG and CP setups were similar, while the reflectance values were higher for the CP setup due to its greater illumination intensity. Reflectance is influenced by the physical structure of soil [35]; the size, shape, and arrangement of particles and voids affect the length of the light transmitted through a soil sample, thereby influencing spectral signatures [40,41]. All spectral signatures followed the typical shape in each wavelength region, i.e., low values in VIS that rise in NIR and SWIR, while showing water absorbance features at around 1400 nm, 1900 nm, and 2200 nm. The 1400–1900 nm absorption bands dominated for water (O–H bonds), even though the peak at 1400 nm was associated with aliphatic C–H and the peak at 1900 nm was associated with amide N–H [15,42]. The spectral shape at 2200 nm was associated with groups such as phenolic O–H, amide N–H, amine N–H, and aliphatic C–H [2]. In sum, the three major reflectance peaks identified were caused by absorbances of O–H bonds of hygroscopically bound water, clay lattices, and various oxides [43].

### 3.2. Laboratory Analysis

Table 2 shows basic statistics for the chemical and physical properties of the soil samples. Since the soil dataset reflected four different locations with different chemical and physical soil properties, values were very diverse. In fact, the coefficients of variation (CoV) obtained for P, Ca, and Fe were large. Generally, the variability observed in the soil samples for some chemical and physical properties was considered appropriate for spectroscopic calibrations, while the variability of other properties (clay and organic matter) was not great enough to build robust PLSR models.

**Table 2.** Descriptive statistics for soil sample properties.

| Soil Property | N | Min | Max | Range | Median | Mean | SD | CoV |
|---|---|---|---|---|---|---|---|---|
| Silt (%) | 48 | 14 | 56 | 42 | 36 | 34.58 | 11.00 | 31.81 |
| Clay (%) | 48 | 10 | 32 | 22 | 19 | 18.46 | 5.91 | 32.02 |
| Sand (%) | 48 | 24 | 76 | 52 | 46 | 46.96 | 15.10 | 32.15 |
| pH | 48 | 5.33 | 8.47 | 3.14 | 7.63 | 7.24 | 1.16 | 16.07 |
| Ec (dS m$^{-1}$) | 48 | 0.02 | 0.12 | 0.10 | 0.08 | 0.07 | 0.03 | 49.27 |
| Organic matter (%) | 48 | 0.37 | 2.40 | 2.03 | 0.81 | 1.02 | 0.52 | 51.04 |
| Total N (%) | 48 | 0.05 | 0.16 | 0.11 | 0.08 | 0.09 | 0.03 | 33.76 |
| P (mg kg$^{-1}$) | 35 | 5.65 | 58.38 | 52.73 | 16.39 | 26.12 | 20.67 | 79.14 |
| K (cmol kg$^{-1}$) | 48 | 0.13 | 0.80 | 0.67 | 0.40 | 0.38 | 0.14 | 37.55 |
| Ca (cmol kg$^{-1}$) | 48 | 1.56 | 20.90 | 19.34 | 9.80 | 10.49 | 6.44 | 61.37 |
| Mn (mg kg$^{-1}$) | 48 | 1.90 | 28.40 | 26.50 | 10.65 | 11.02 | 7.09 | 64.35 |
| Fe (mg kg$^{-1}$) | 48 | 2.56 | 212.41 | 209.85 | 8.00 | 28.60 | 41.49 | 145.11 |

An important issue in chemometric calibration is collinearity in the analytical values conditioning the validity of the results [23]. Pearson correlations (not shown) were calculated for all the soil properties; the greatest correlations were observed for silt with sand (r = 0.95) and for pH with Ec (r = 0.94), while the lowest correlation was found for Fe with K.

### 3.3. PLSR Model Predictions

Only variables with $R^2$ values above 0.60 were considered for PLSR in this research. The reference for the other preprocessing results was PLSR prediction results using the full VIS + NIR + SWIR range (350–2500 nm) and non-preprocessed data, as summarised in Table 3. Broadly speaking, the PLSR calibration results indicated good predictions for pH, Ec, and P, and reasonably good predictions for sand, N, K, and Mn. The CP setup models had higher $R^2$ and lower RMSE values for pH, Ec, P, Ca, and Mn, while the PG setup models had higher $R^2$ and lower RMSE values for sand, N, and K. Regarding the number of factors, CP setup models required fewer factors that PG setup models.

**Table 3.** CV statistics for PLSR for the full VIS + NIR + SWIR range (350–2500 nm).

| Soil Property | PG Setup | | | | | CP Setup | | | | |
|---|---|---|---|---|---|---|---|---|---|---|
| | $R^2$ | RMSE | SE | RPD | Factors | $R^2$ | RMSE | SE | RPD | Factors |
| Sand | 0.75 | 7.678 | 7.759 | 1.95 | 7 | 0.70 | 8.327 | 8.415 | 1.79 | 6 |
| pH | 0.95 | 0.340 | 0.343 | 3.38 | 4 | 0.92 | 0.329 | 0.334 | 3.47 | 4 |
| Ec | 0.89 | 0.011 | 0.011 | 2.73 | 3 | 0.90 | 0.011 | 0.011 | 2.73 | 4 |
| N | 0.68 | 0.017 | 0.017 | 1.76 | 6 | 0.62 | 0.018 | 0.018 | 1.67 | 3 |
| P | 0.90 | 6.530 | 6.619 | 3.09 | 7 | 0.90 | 6.647 | 6.741 | 3.04 | 4 |
| K | 0.65 | 0.086 | 0.087 | 1.61 | 6 | 0.64 | 0.087 | 0.088 | 1.59 | 6 |
| Ca | 0.87 | 2.332 | 2.357 | 2.73 | 6 | 0.89 | 2.141 | 2.163 | 2.98 | 6 |
| Mn | 0.62 | 4.399 | 4.446 | 1.59 | 3 | 0.66 | 4.195 | 4.239 | 1.67 | 5 |

The PLSR predictions explained about 90% of the variance in the laboratory analyses of pH, Ec, and P, with RMSE values of 0.340–0.329, 0.01–0.01 dS m$^{-1}$, and 6.530–5.770 mg kg$^{-1}$, respectively, for the PG and CP setups. For sand, N, and K, $R^2$ values were better for the PG setup (0.75, 0.68, and 0.65, respectively) than for the CP setup (0.70, 0.62, and 0.64, respectively). RMSE results were very similar; the most different were predictions for sand: 7.678% for the PG setup and 8.327% for the CP setup. Predictions for Mn ranged from $R^2$ = 0.62 (RMSE = 4.399 mg kg$^{-1}$) for the PG setup to $R^2$ = 0.66 (RMSE = 4.195 mg kg$^{-1}$) for the CP setup.

Cozzolino and Morón [2] suggest that calibration models developed for soil composition by spectroscopy can be classified according to RPD as poor (<1.6), acceptable (1.6–2.0), or excellent (>2.0). According to this classification, the fitted PLSR models proved excellent

for pH, P, and Ca and acceptable for sand, Ec, N, K, and Mn. Chang et al. [35] suggest that spectroscopic prediction models in the intermediate category could be improved using different calibration strategies. The strategy used in this research was SVN and DT preprocessing to achieve models that reduce errors (RMSE and SE) and number of factors and increase $R^2$.

Table 4 shows PLSR results for preprocessed reflectance. Regarding the SVN transformation, $R^2$ increased for all variables except for P, Ec, and N. PLSR performance improved more for the CP setup models. RMSE decreased except for N and K, while the number of factors was also reduced except for N. Regarding DT preprocessing, $R^2$ did not increase except for K. Results were generally better for the PG setup models. RMSE values were maintained or increased except for P and Mn, while the number of factors was reduced except for N. The reduction in the number of factors was less for PG setup models. For SVN preprocessing, $R^2$ values increased except for Ec and pH, which remained constant, while RMSE values decreased. Results were better for the models based on the CP setup, while the number of factors was also reduced, with the exception of the PG models estimating Mn (+2 factors) and the CP setup models estimating N and P (+1 factor). For DT preprocessing, although not significantly greater, $R^2$ and RMSE values were better for the CP setup than the PG setup. The main improvement was the simplification of the models in reducing the number of factors. Finally, applying SVN + DT, $R^2$ values increased and RMSE values decreased, while the number of factors was reduced, with the exception of N (7 factors).

**Table 4.** CV statistics for PLSR for the full VIS + NIR + SWIR range (350–2500 nm) and spectral transformations.

| | PG Setup | | | | | CP Setup | | | | |
|---|---|---|---|---|---|---|---|---|---|---|
| Soil Property | $R^2$ | RMSE | SE | RPD | Factors | $R^2$ | RMSE | SE | RPD | Factors |
| **SVN** | | | | | | | | | | |
| Sand | 0.76 | 7.556 | 7.635 | 1.98 | 6 | 0.72 | 8.028 | 8.112 | 1.86 | 5 |
| pH | 0.96 | 0.317 | 0.320 | 3.63 | 4 | 0.92 | 0.336 | 0.339 | 3.42 | 3 |
| Ec | 0.89 | 0.011 | 0.011 | 2.73 | 2 | 0.91 | 0.010 | 0.011 | 2.73 | 3 |
| N | 0.70 | 0.016 | 0.016 | 1.88 | 5 | 0.65 | 0.018 | 0.018 | 1.67 | 4 |
| P | 0.92 | 5.891 | 5.977 | 3.43 | 5 | 0.93 | 5.701 | 5.784 | 3.54 | 5 |
| K | 0.66 | 0.084 | 0.085 | 1.65 | 5 | 0.66 | 0.084 | 0.085 | 1.65 | 5 |
| Ca | 0.89 | 2.119 | 2.141 | 3.01 | 5 | 0.91 | 1.960 | 1.981 | 3.25 | 5 |
| Mn | 0.65 | 4.115 | 4.154 | 1.71 | 5 | 0.69 | 3.992 | 4.032 | 1.76 | 5 |
| **DT** | | | | | | | | | | |
| Sand | 0.74 | 7.725 | 7.805 | 1.93 | 4 | 0.73 | 7.902 | 7.985 | 1.89 | 7 |
| pH | 0.91 | 0.356 | 0.359 | 3.23 | 3 | 0.92 | 0.343 | 0.346 | 3.35 | 3 |
| Ec | 0.90 | 0.011 | 0.011 | 2.73 | 3 | 0.90 | 0.007 | 0.011 | 2.73 | 3 |
| N | 0.68 | 0.017 | 0.017 | 1.76 | 4 | 0.67 | 0.017 | 0.017 | 1.76 | 3 |
| P | 0.91 | 6.307 | 6.398 | 3.20 | 4 | 0.93 | 5.719 | 5.795 | 3.53 | 4 |
| K | 0.68 | 0.082 | 0.083 | 1.69 | 4 | 0.64 | 0.087 | 0.087 | 1.44 | 4 |
| Ca | 0.86 | 2.412 | 2.437 | 2.64 | 4 | 0.90 | 2.043 | 2.064 | 3.12 | 5 |
| Mn | 0.68 | 4.075 | 4.118 | 1.72 | 4 | 0.64 | 4.315 | 4.360 | 1.63 | 1 |
| **SVN + DT** | | | | | | | | | | |
| Sand | 0.77 | 7.280 | 7.74 | 1.95 | 6 | 0.76 | 7.550 | 7.62 | 1.98 | 6 |
| pH | 0.93 | 0.315 | 0.32 | 3.64 | 4 | 0.92 | 0.342 | 0.35 | 3.35 | 3 |
| Ec | 0.90 | 0.011 | 0.01 | 2.73 | 2 | 0.91 | 0.011 | 0.01 | 2.73 | 3 |
| N | 0.71 | 0.016 | 0.02 | 1.88 | 7 | 0.70 | 0.016 | 0.02 | 1.88 | 7 |
| P | 0.92 | 6.083 | 6.17 | 3.32 | 4 | 0.92 | 5.900 | 5.99 | 3.42 | 4 |
| K | 0.68 | 0.083 | 0.08 | 1.69 | 6 | 0.63 | 0.088 | 0.09 | 1.57 | 4 |
| Ca | 0.89 | 2.198 | 2.22 | 2.90 | 4 | 0.91 | 2.005 | 2.03 | 3.18 | 4 |
| Mn | 0.71 | 3.887 | 3.93 | 1.80 | 5 | 0.69 | 3.983 | 4.03 | 1.76 | 5 |

Preprocessing: SVN: standard normal variate; DT: detrending transformation; SVN + DT: standard normal variate plus detrending transformation.

## 4. Discussion

### 4.1. Soil Property Predictions

Cross-validation results for the PLSR models were different for the three particle-size distributions (clay, silt, and sand). For sand, results were satisfactory ($R^2 = 0.75$ and $R^2 = 0.70$ for the PG and CP data, respectively), and also corroborated other published results [17,39,44]. For clay, however, results were quite poor ($R^2 = 0.53$ and $R^2 = 0.51$ for the PG and CP data, respectively), and likewise for silt ($R^2 = 0.51$ and $R^2 = 0.49$ for the PG and CP data, respectively). Those unexpectedly low $R^2$ values may be due to narrow variability in clay content (min = 10 and max = 32) and silt content (min = 14 and max = 42) of the analysed soils.

Soil content in N was estimated by spectroscopy because it is quite sensitive to IR radiation. The $R^2$ values obtained ranged from $R^2 = 0.68$ (RMSE = 0.017%) to $R^2 = 0.62$ (RMSE = 0.018%) for the PG and CP setups, respectively, lower than the values of $R^2 = 0.80$–0.98 cited elsewhere [45] and the $R^2 = 0.92$ (SE = 2.19) obtained by Cozzolino et al. [23]. Our result can be explained by the fact that N estimation by spectroscopy is soil-dependent, due mainly to varying carbonate contents [46]. While MIR-ATR spectroscopy can, in fact, predict nitrate concentration in soil pastes by direct measurement, prediction accuracy is strongly conditioned for water and soil constituents [47–49].

Previous research has reported the ability of soil reflectance spectroscopy to accurately determine soil organic matter [24]. However, our results for organic matter were $R^2 = 0.29$ (RMSE = 0.438%) and $R^2 = 0.27$ (RMSE = 0.445%) for the PG and CP setups, respectively. These poor results may be due to the fact that the analysed soils have low organic matter content (0.37–2.40%) and high sand content (24–76%). In fact, spectroscopic predictions of organic matter are poorly accurate in soils with low C content [50] and high sand content [14].

Soil pH is a key factor for agriculture as an important fertility regulator of nutrient solubility and plant root development, biological activity, decomposition, mineralisation, etc. Because pH is a soil property with no direct spectral responses in the NIR spectroscopy range [15], calibrations rarely perform better than an RMSE of one-third or half a pH unit [14]. However, soil pH has been predicted quite successfully in several studies [19,51,52].

The pH prediction performance of our PLSR models was excellent (RPD > 3.3) for both PG and CP reflectances ($R^2 = 0.92$ (RMSE = 0.340) and $R^2 = 0.92$ (RMSE = 0.329), respectively). Our results were similar to those reported by Kuang et al. (RMSE = 0.36; RPD = 2.02) and better than those reported by Sorenson et al. [52] ($R^2 = 0.68$) and by Marín-González et al. [19] ($R^2 = 0.86$; RPD = 2.69). Results for pH predictions may be explained by co-variation to spectrally active soil constituents such as organic matter and clay [35] or by soil mineralogy and carbonate content [15]. Note that pH calibrations tend to vary from one dataset to another because they reflect different scenarios.

Our results for Ca calibrations were reasonably good, at $R^2 = 0.87$ (RMSE = 2.332 cmol kg$^{-1}$) and $R^2 = 0.88$ (RMSE = 2.141 cmol kg$^{-1}$) for the PG and CP setups, respectively, while RPD > 2 indicated excellent accuracy. Those results improved on those obtained elsewhere: $R^2 = 0.75$ (RMSE = 4.00 cmol kg$^{-1}$) by Chang et al. [35]; $R^2 = 0.72$ (RMSE = 4.20 cmol kg$^{-1}$) by Islam et al. [6]; and $R^2 = 0.67$ (RMSE = 3.89 cmol kg$^{-1}$) by Soriano-Disla et al. [53]. Similar results to ours were obtained by Shepherd and Walsh [54] ($R^2 = 0.88$), Cozzolino and Morón [2] ($R^2 = 0.90$), and Marín-González et al. [19] ($R^2 = 0.89$; RMSE = 22.05).

As for Ec, the results of the PLSR models indicate accurate predictions for both the PG setup ($R^2 = 0.89$; RMSE = 0.011; RPD = 2.73) and the CP setup ($R^2 = 0.90$; RMSE = 0.011; RPD = 2.73). Our results corroborate those of Lei et al. [55], who fitted a PLSR model with $R^2 = 0.69$ (RMSE = 0.039) for semi-arid grasslands, and Farifteh et al. [56], who fitted models that ranged from $R^2 = 0.80$ (RMSE = 0.070) to $R^2 = 0.80$ (RMSE = 0.060) for sandy and agricultural areas, while Mashimbye et al. [57] reported PLSR models with $R^2 = 0.65$–0.85 (validation) for soil in South Africa.

According to Marín-González et al. [19], K is a difficult property to estimate with NIR spectroscopy. Islam et al. [6] reported poor predictions for extractable K for calibrations

($R^2$ = 0.29) using PCR. Using PLSR, Volkan et al. [18] reported K soil content estimates (depending on the validation set) with $R^2$ values between 0.32 and 0.25 (RMSE 0.21 and 0.22, respectively). Our models predicted extractable K with moderate accuracy for both the PG setup ($R^2$ = 0.65; RMSE = 0.086; RPD = 1.61) and the CP setup ($R^2$ = 0.64; RMSE = 0.088; RPD = 1.59). Those accuracy levels corroborate Zornoza et al. [42], who reported PLSR models with $R^2$ = 0.79 (RMSE = 0.11 g kg$^{-1}$; RPD = 2.19), but using soil with high K content (mean 0.60 g kg$^{-1}$).

Our extractable Mn estimates were moderately accurate for both PG ($R^2$ = 0.62; RMSE = 4.399 mg kg$^{-1}$; RPD = 1.59) and CP ($R^2$ = 0.66; RMSE = 4.195 mg kg$^{-1}$; RPD = 1.67) setups. There are few references in the literature for K content estimations using VIR-NIR, although Chang et al. [35] built a PCR model resulting in $R^2$ = 0.70 (RMSE = 56.40 mg kg$^{-1}$; RPD = 1.79).

Since our PG and CP setup models had comparable predictive capacities ($R^2$) and accuracies (RMSE), it can be concluded that the CP setup, more versatile for field measurements, is preferable for soil property estimates by VIS-NIR-SWIR spectroscopy. Note that while Rosero-Blasova et al. [33] reported a PG setup to perform better than a CP setup, they detected no statistically significant differences between the two setups.

### 4.2. PLSR Model Performance

Figure 2 shows distributions of the weighted regression coefficients over the full spectral range for both the PG and CP setups and the considered soil properties (to highlight differences, the regression coefficients for each soil property are offset by 3.0 units). Evident are several peaks in wavelength bands located in the VIS and NIR regions, attributable to colour, water, organic matter, and clay minerals [16]. Regarding sand, K, P, and Mn, the main peaks in the VIS range are associated with the blue and green regions around 450 nm and 550 nm, respectively, demonstrating that colour contributes similarly to predicting those properties. Mouazen et al. [16] reported a similar distribution of regression coefficients to ours, identifying the spectral range between 1800 nm and 2450 nm as the most active for P and K estimates. As for pH and Ec, these are mainly associated with the blue and green regions, denoting the influence of Fe oxides associated with clay minerals [58]. Predictions of N content are little affected by colour, while Ca predictions are influenced in the red region.

As was expected, regression coefficient distributions were very similar for both PG and CP setups, thereby corroborating measurement and prediction consistency between both. The main difference was reported for N estimates, which can be attributed to PLSR algorithm calculations instead of actual differences in spectral signatures.

Figure 3 shows the $R^2$ values for the PLSR models obtained for the different data subsets, different pre-processing approaches, and PG and CP setups. Using only part of the spectrum (VIS, NIR, or SWIR), general trends for $R^2$ were similar: $R^2$ values with SWIR were the best, followed by $R^2$ values with VIS, then $R^2$ values with NIR. RMSE values were highest with NIR, while the lowest values were obtained with SWIR (see Table 4 above). The main disadvantage of using SWIR subsets was that some models (e.g., those for N and Mn) needed a greater number of factors than the VIS and NIR subsets, which potentially delays computational calculations.

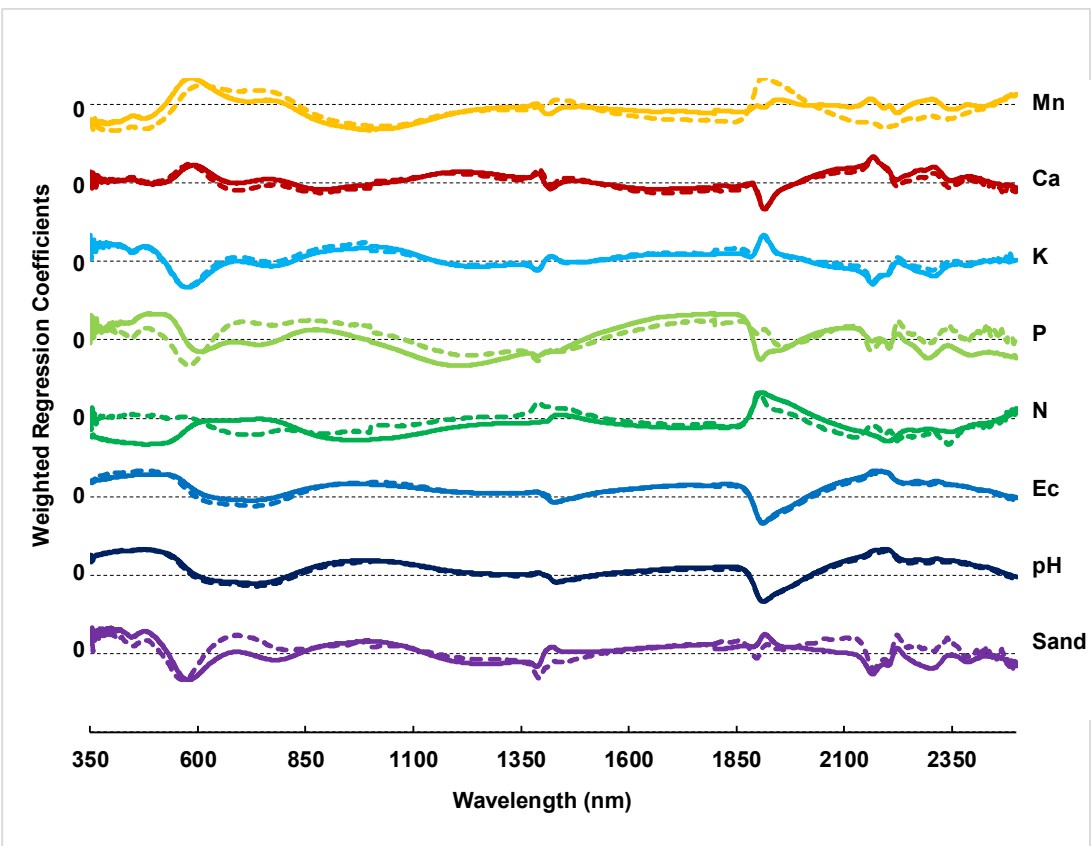

**Figure 2.** Weighted regression coefficient distribution over the spectral range obtained for PLSR models for the CP setup (coloured unbroken lines) and PG setup (coloured broken lines). Soil properties analysed for cross-validation are sand content (Sand), pH, electrical conductivity (Ec), total nitrogen content (N), extractable phosphorous (P), extractable potassium (K), extractable calcium (Ca), and extractable manganese (Mn). Black broken lines represent zero correlation, offset by 3.0 units for clarity of presentation.

### 4.3. Data Preprocessing

Table 5 shows the best fitting models for each soil property, each type of preprocessing, each spectral subset, and each setup. Generally, sand, pH, Ec, N, P, and K were best predicted with models using the SWIR subset, Ca with models using the VIS subset, and Mn with models using the full spectrum. The best models for sand, Ec, K, Ca, and Mn were obtained for SVN + DT preprocessing, while models for N and P only required SVN preprocessing, and models for pH required no preprocessing.



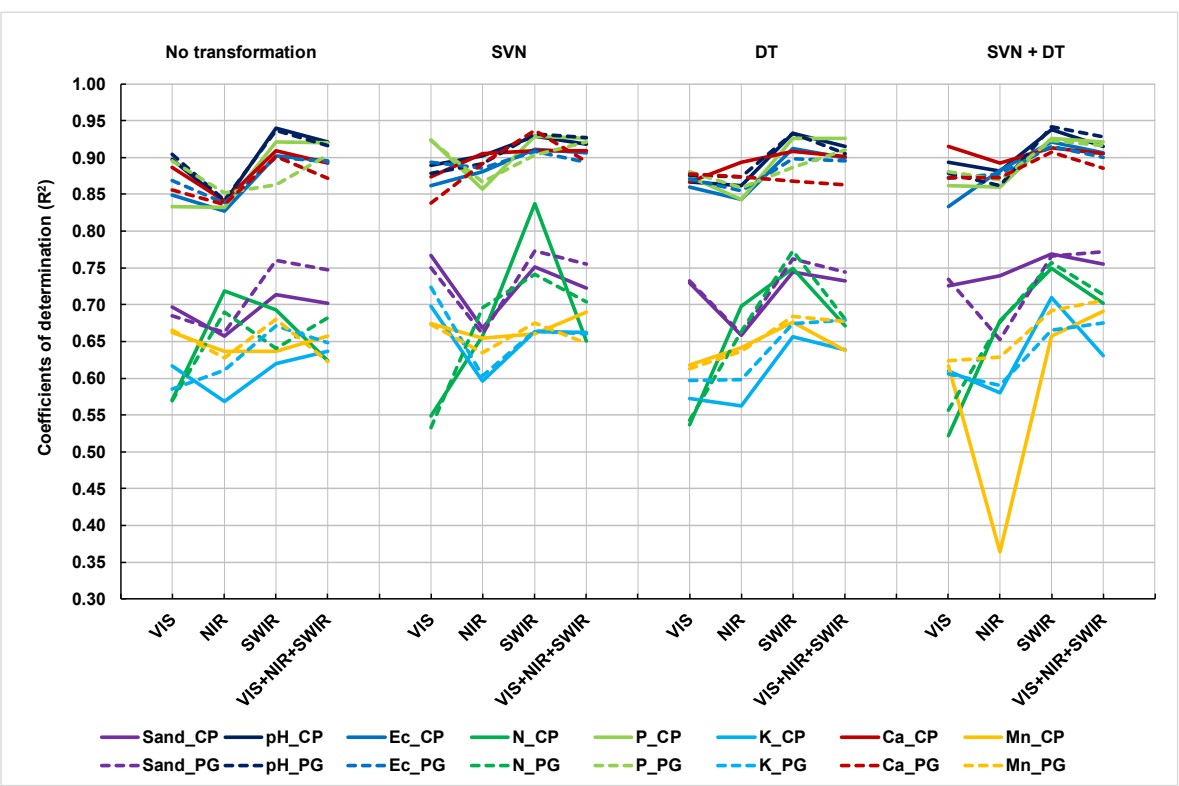

**Figure 3.** Variation in coefficients of determination for spectral subsets and preprocessing approaches. $R^2$ values obtained for PLSR models for the CP setup (coloured unbroken lines) and PG setup (coloured broken lines). Spectral subsets were VIS (350–700 nm), NIR (701–1000 nm), SWIR (1001–2500 nm), and VIS + NIR + SWIR (350–2500 nm). Spectral preprocessing approaches were standard normal variate (SVN), detrending transformation (DT), and SVN + DT. Soil properties are sand content (Sand), pH, electrical conductivity (Ec), total nitrogen content (N), extractable phosphorous (P), extractable potassium (K), extractable calcium (Ca), and extractable manganese (Mn).

**Table 5.** Best PLSR models for predicting soil properties.

| Soil Property | Transformation | Spectral Subset | PG Setup | | | | | CP Setup | | | | |
| --- | --- | --- | --- | --- | --- | --- | --- | --- | --- | --- | --- | --- |
| | | | $R^2$ | RMSE | SE | RPD | Factors | $R^2$ | RMSE | SE | RPD | Factors |
| Sand | SVN + DT | SWIR | 0.77 | 7.276 | 7.353 | 2.05 | 6 | 0.77 | 7.326 | 7.396 | 2.04 | 6 |
| pH | None | SWIR | 0.94 | 0.284 | 0.287 | 4.04 | 3 | 0.94 | 0.287 | 0.29 | 4.00 | 5 |
| Ec | SVN + DT | SWIR | 0.92 | 0.010 | 0.01 | 3.00 | 3 | 0.92 | 0.010 | 0.01 | 3.00 | 3 |
| N | SVN | SWIR | 0.77 | 0.014 | 0.014 | 2.14 | 7 | 0.84 | 0.012 | 0.016 | 1.88 | 7 |
| P | SVN | SWIR | 0.92 | 5.816 | 5.891 | 3.48 | 5 | 0.93 | 5.571 | 5.653 | 3.62 | 5 |
| K | SVN + DT | SWIR | 0.72 | 0.076 | 0.077 | 1.82 | 4 | 0.71 | 0.078 | 0.079 | 1.77 | 4 |
| Ca | SVN + DT | VIS | 0.94 | 1.603 | 1.983 | 3.25 | 5 | 0.92 | 1.859 | 2.277 | 2.83 | 5 |
| Mn | SVN + DT | VIS + NIR + SWIR | 0.71 | 3.887 | 3.928 | 1.80 | 5 | 0.69 | 3.983 | 4.025 | 1.76 | 5 |

Transformations: SVN: standard normal variate; DT: detrending, SVN + DT: standard normal variate plus detrending. Spectral subsets: VIS (350–700 nm), SWIR (1001–2500 nm), VIS + NIR + SWIR (350–2500 nm).

No one specific kind of preprocessing ensures the effectiveness of models. Spectral signatures of soils are influenced by chemical composition and structural properties that produce non-linear light scattering effects. Regression model performance depends on the soil dataset, the analysed soil property, and the variability of the data [59], so a specific model needs to be fitted that reflects each scenario. Furthermore, it has been reported that spectral preprocessing has a minor influence on results when PLSR models are used [60].

Stenberg et al. [14] report that SVN combined with DT is one of the more commonly used means of improving PLSR performance, as this approach usually enhances weak soil spectral signals. In our research, while SVN + DT increased $R^2$ and reduced RMSE (see Tables 3 and 4), improvement depended on the studied soil property, and was not so great probably because the raw reflectance data were quite stable and consistent. Other

authors [33,61] report, for VIS-NIR spectroscopy, that preprocessing of spectral samples is data-specific, so no single or combination technique is generally applicable to preprocessing. In fact, different preprocessing methods should be used for different calibration techniques, different datasets, and different soil conditions [59].

Table 5 confirms that the predictive performance of soil property PLSR spectroscopic models varies with different kinds of preprocessing. Furthermore, use of different accessories results in different illumination setups and observation geometries that condition measurement and that consequently may affect the performance of models [21]. Model effectiveness is also probably conditioned by variability in the data [59]. In fact, for properties where standard deviations are greater, more variance is explained and greater accuracy is achieved.

Figure 4 represents Pearson coefficient values reflecting correlations between soil properties and wavelengths. The correlograms grouped by correlation structure are Fe and Mn; N and organic matter; pH and Ec; and sand and K. Analysing the correlograms, within groups, the correlation structure is quite redundant; only Fe and organic matter have direct optical features, while predictions for the remaining properties are based on spurious correlations [62]. Patterns in the groups can be attributed to dominant chemical characteristics (e.g., iron oxides and clay minerals in the group consisting of Fe and Mn) and to the aggregate effect of several optically active minerals [62]. Regression models based on spurious correlations depend on underlying geology and soil parameters and so only are useful for our studied plots.

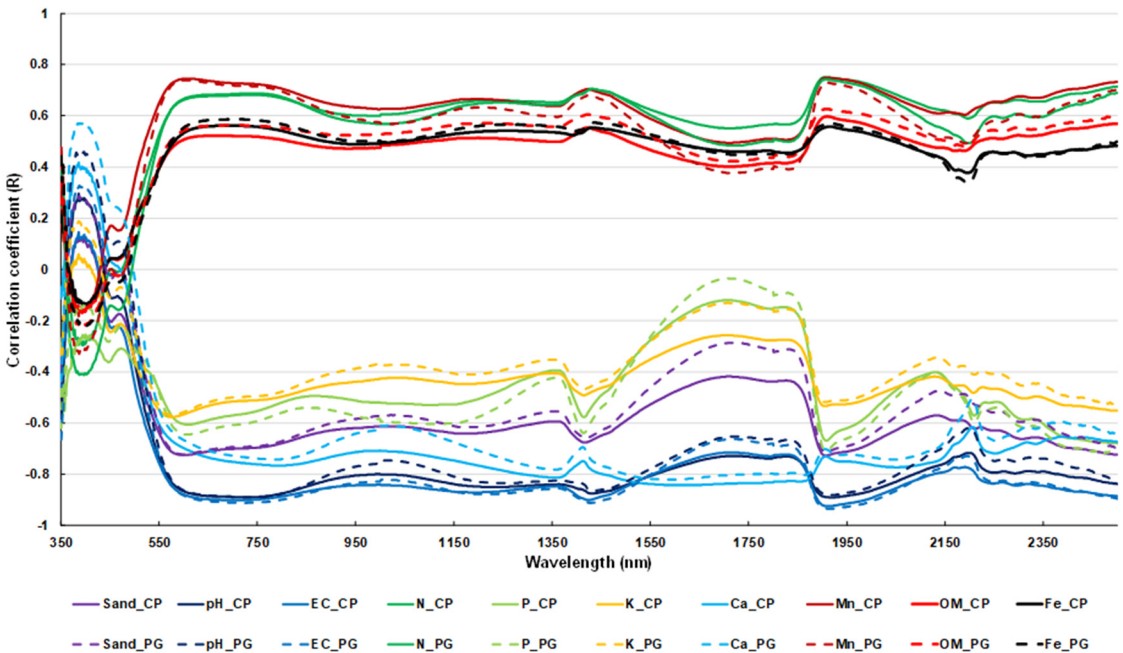

**Figure 4.** Correlograms of soil properties obtained from simple linear correlations for each wavelength for the CP setup (coloured unbroken lines) and the PG setup (coloured broken lines). Soil properties are sand content (Sand), pH, electrical conductivity (Ec), total nitrogen content (N), extractable phosphorous (P), extractable potassium (K), extractable calcium (Ca), extractable manganese (Mn), organic matter (OM), and extractable iron (Fe).

Interpretation of the regression coefficient curves and correlograms (Figures 3 and 4, respectively) is complicated due to the complexity in overlapping soil constituent absorption patterns. The studied chemical properties do not have direct spectral responses in the considered spectral regions. The prediction of these properties, namely, sand, pH, Ec, N, P, K, Ca, and Mn, can be attributed to locally present co-variations in spectrally active constituents (mainly organic C and clay minerals). Furthermore, correlations of some soil properties with NIR spectroscopy are still unknown and so require further investigation [16]. In fact, Miller [63] acknowledged that it is difficult to identify relevant effects

in the NIR spectrum based on chemistry and spectroscopy of samples alone. Therefore, further studies are needed to understand why, in our study and using VIS-NIR-SWIR spectroscopy, properties were estimated with excellent accuracy (pH, Ec, P and Ca) and acceptable accuracy (sand, N, and Mn).

Our results suggest that it is possible to estimate variables such as sand, pH, Ec, N, P, K, Ca, and Mn that are optically non-active chemical properties with featureless spectra, because those elements are bonded to spectrally active soil components, mainly iron oxides, organic matter, and clay minerals, in such a way that the bonds constitute a key predictive mechanism [62]. Similar conclusions have been published by Martínez-Carreras et al. [64] and Wu et al. [65].

## 5. Conclusions

Vineyard soil parameters were calculated by relating spectral signatures and laboratory analytical determinations using PLSR. Reflectance measurements were made using PG and CP setups. Our findings suggest that proximal soil spectroscopy is a useful technique for soil characterisation and monitoring. The great advantage of the spectroscopic approach is that it is cost-effective and rapid, although prediction accuracy is less than for laboratory analyses. The predictive capacity ($R^2$) and accuracy (RMSE) of the PLSR models depends on setup (PG or CP), preprocessing (SVN and/or DT), spectral subset (VIS, NIR, SWIR, or full spectrum), and individual soil properties. The best predictions, with $R^2$ values above 0.915, were obtained for pH, Ec, and P, while moderately accurate predictions, with $R^2$ values of 0.69 to 0.77, were obtained for sand, N, and K.

In conclusion, PLSR models can be useful for monitoring overall changes in soil properties. Further studies aimed at more effective precision viticulture practices will focus on vineyard soil characterisation using VIS-NIR-SWIR spectroscopy combined with geographical information system (GIS) data.

**Author Contributions:** Conceptualisation, M.G.-F., V.M. and J.R.R.-P.; methodology, M.G.-F., V.M. and J.R.R.-P.; software, E.S.-A.; formal analysis, V.M., E.S.-A. and J.R.R.-P.; investigation, D.P.-O.; writing—original draft preparation, M.G.-F., D.P.-O. and J.R.R.-P.; writing—review and editing, D.P-O. and J.R.R.-P.; supervision and project administration, J.R.R.-P. All authors have read and agreed to the published version of the manuscript.

**Funding:** This research was funded by the Education Department of the Junta de Castilla y León, grant number LE112G18.

**Institutional Review Board Statement:** Not applicable.

**Informed Consent Statement:** Not applicable.

**Data Availability Statement:** Not applicable.

**Acknowledgments:** This research was supported by funding from the Education Department of the Junta de Castilla y León-Spain (grant number LE112G18) under a call for research by recognised research groups attached to public universities starting in 2018 (Order 20 November 2017). Marta García Fernández gratefully acknowledges financial support provided by the Fundación Carolina Rodríguez and the Universidad de León.

**Conflicts of Interest:** The authors declare no conflict of interest. The authors declare that they have no known competing financial interests or personal relationships that could have appeared to influence the work reported in this paper.

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
