# Peer review of "Estimating Soil Properties and Nutrients by Visible and Infrared Diffuse Reflectance Spectroscopy to Characterize Vineyards"

_agronomy, doi:10.3390/agronomy11101895_

Round 1

Reviewer 1 Report

I have already reviewed this manuscript as reference agronomy-1312146 Estimating soil properties and nutrients by VIS-NIR-SWIR re- flectance spectroscopy to characterize vineyards, and after checking the changes made by the authors I have to state that the summary has been improved, the title has been changed, the experimental design has been rewritten; in the methodology and references they have followed most of the recommendations given. Therefore, as a consequence of the changes and corrections incorporated in this new version, the scientific quality of the manuscript has considerably improved..

Reviewer 2 Report

I have already reviewed this manuscript as reference agronomy-1312146 Estimating soil properties and nutrients by VIS-NIR-SWIR re- flectance spectroscopy to characterize vineyards, and after checking the changes made by the authors I have to state that the summary has been improved, the title has been changed, the experimental design has been rewritten; in the methodology and references they have followed most of the recommendations given. Therefore, as a consequence of the changes and corrections incorporated in this new version, the scientific quality of the manuscript has considerably improved.

This manuscript is a resubmission of an earlier submission. The following is a list of the peer review reports and author responses from that submission.

Round 1

Reviewer 1 Report

Using a portable spectroradiometer and compared for pistol grip versus contact probe setups, the authors documents the importance of visible, near, and shortwave infrared reflectance spectroscopy, to predict soil sample properties for vineyards. It is an interesting paper of high importance because reflectance spectroscopy is expected to contribute as a technique for characterizing soils for precision viticulture purposes. The article is well written for the most part and should be of interest to readers. The results are consistent with the proposed objectives.   I have only some suggestion:

- It does not seem reasonable to use acronyms in the title as VIS-NIR-SWIR

-Line 38. It is true that “Assessment of soils using visible (VIS), near infrared (NIR), and shortwave infrared (SWIR) spectroscopy has been shown to be a fast, cost-effective, environmentally friendly, nondestructive, reproducible, and repeatable analytical technique [4]. It is also easy to use since samples only require minimal preparation, and, furthermore, requires no chemicals and reagents and so does not generate chemical waste [5]” but to the date it has not served to replace laboratory methods; or does you think so.

-Line 107 “Spectroscopy has previously been applied to viticulture. For vineyards located in Australia, Cozzolino et al. [23] evaluated use of a portable NIR spectrophotometer in the field to predict soil chemical properties, fitting PLSR models with coefficients of determination (R2 ) that ranged from 0.69 for P to 0.95 for total N content”. What does you bring like new, where does the difference lie with this job?

-Lithic Leptosol and Albic Arenosol are not common soils in vineyards, why have you chosen these soils?

-Line 147. Instrumental Techniques Laboratory attached to León University (certified by UNE-EN ISO 9001) following recognized analytical methods [30]. This is very generic. Instead, from line 150 to 22 is dedicated to document the application technique of visible, near, and shortwave infrared reflectance spectroscopy. Justify.

-Line 187. The key to this research is that PLSR was used to estimate soil parameters (predicted variables) from spectral signatures (predictor variables). It would be convenient to explain the rationale and how to go from this procedure to obtaining the data itself.

-Line 309 Cross-validation results for the PLSR models were different for the three soil textures. For sand, results were satisfactory (R2=0.747 and R2=0.702 for the PG and CP data, respectively), and also corroborated other published results [17,38,43]. For clay, however, results were quite poor (R2=0.534 and R2=0.505 for the PG and CP data, respectively), and likewise for silt (R2=0.504 and R2=0.494 for the PG and CP data, respectively). Based on this sentence, it is possible to wonder if the method is valid or not

-Line 477 Our results suggest that it is possible to estimate variables such as sand, pH, Ec, N, P, K, Ca, and Mn that are optically nonactive chemical properties with featureless spectra, because those elements are bonded to spectrally active soil components, mainly iron oxides, organic matter and clay minerals, in such a way that the bonds constitute a key predictive mechanism [61]. Based on this sentence, it is possible to wonder if the method is valid or not

-Line 500 PLSR models can be useful for monitoring overall soil changes and generating soil maps. In my opinion this sentence is speculative. Making a soil map is much more complex than this.

-Reference 29. IUSS Working Group WRB. World Reference Base for Soil Resources 2014: Update 2015 International Soil Classification System for Naming Soils and Creating Legends for Soil Maps.; World Soil Resources Reports; FAO: Rome, 2015; ISBN 978- 595 92-5-108369-7. Please recommended citation: IUSS Working Group WRB. 2015. World Reference Base for Soil Resources 2014, update 2015 International soil classification system for naming soils and creating legends for soil maps. World Soil Resources Reports No. 106. FAO, Rome

I wish those changes will contribute to improve your paper.

Author Response

Estimating soil properties and nutrients by visible and infrared diffuse reflectance spectroscopy to characterize vineyards

Ref.: Agronomy-1312146

Revision #1

Response to Reviewer #1‘s comments

REVIEWER #1

Before responding to your comments and suggestions, we’d like to thank you for your review of this work, which has contributed greatly to improving this article and to the focus of future work.

For point-by-point response, please see attached file.

Reviewer 2 Report

The manuscript titled “Estimating soil properties and nutrients by VIS-NIR-SWIR reflectance spectroscopy to characterize vineyards” report some results on the ability of reflectance spectroscopy in estimating soil properties and nutrients. Authors collected 48 soil samples that were subjected to direct (laboratory analyses) and indirect (reflectance spectroscopy) to test two measurement methods and two modelling approaches.  Despite the study is interesting,  authors should clearly indicate what are the new findings compared to previous works on the same topic.

Title must be shortened as follow: “Estimating soil properties and nutrients by VIS-NIR-SWIR reflectance spectroscopy”. The results of this study can be transferred to any crop soil. The only link with the grapevine is that soil samples were collected in vineayards.

Line 125: add references and specify differences with this study.

Fig.1 indicate the soil samples on which the spectral signatures were measured

Fig. 2: the y-axis must be added

Fig.3 : R2 values at different spectral bands are represented as continuum lines, but they do not represent any course. These data should be presented as a table

Three or two decimals are used for R2 values. Please, use the same number of decimals thorough the manuscript.

Line 249: "denoting different viticulture practice"must be removed, too speculative. Authors don’t know (they didn’t report information about that) whether these differences are caused by viticultural practices or they are intrinsic differences.

Author Response

Estimating soil properties and nutrients by visible and infrared diffuse reflectance spectroscopy to characterize vineyards

Ref.: Agronomy-1312146

Revision #1

Response to reviewer#2’s comments

Reviewer 2

Thanks for your review of this work. Your comments were very appropriate and have contributed greatly to improving this article and to the focus of future work.

For point-by-point response, please see attached file.
